# Subsequent Domino Osteoporotic Vertebral Fractures Adversely Affect Short-Term Health-Related Quality of Life: A Prospective Multicenter Study

**DOI:** 10.3390/medicina59030590

**Published:** 2023-03-16

**Authors:** Tomoyuki Kusukawa, Keishi Maruo, Masakazu Toi, Tetsuto Yamaura, Masaru Hatano, Kazuma Nagao, Hayato Oishi, Yutaka Horinouchi, Fumihiro Arizumi, Kazuya Kishima, Norichika Yoshie, Toshiya Tachibana

**Affiliations:** 1Department of Orthopaedic Surgery, Hyogo Medical University, Nishinomiya 663-8131, Japan; 2Department of Orthopaedic Surgery, Miyoshi Hospital, Miyoshi 778-0005, Japan; 3Department of Orthopaedic Surgery, Daiwa Central Hospital, Osaka 557-0025, Japan; 4Department of Orthopaedic Surgery, Goushi Hospital, Nagasu Nishidori 660-0807, Japan; 5Department of Orthopaedic Surgery, Harima Hospital, Asahi Aioi 678-0031, Japan; 6Department of Orthopaedic Surgery, Osaka Minato Central Hospital, Osaka 552-0003, Japan; 7Department of Orthopaedic Surgery, Takarazuka City Hospital, Takarazuka 665-0827, Japan; 8Department of Orthopaedic Surgery, Sasayama Medical Center, Hyogo Medical University, Tamba-Sasayama 669-2321, Japan

**Keywords:** domino osteoporotic vertebral fractures, magnetic resonance imaging, quality of life, osteoporosis, conservative treatment

## Abstract

*Background and Objectives*: Conservative treatment is the gold standard for acute osteoporotic vertebral fractures (AOVFs). However, the treatment strategy for multiple AOVFs remains unknown. We conducted a prospective study using magnetic resonance imaging (MRI) to investigate how rapidly subsequent osteoporotic vertebral fractures (OVFs) occur as domino OVFs within 3 months. This study aimed to assess the incidence and impact of domino OVFs on quality of life (QOL) following conservative treatment for initial AOVFs. *Materials and Methods*: A prospective multicenter cohort study was conducted at eight hospitals. The included patients were those with AOVFs occurring within 3 weeks, aged >60 years, and diagnosed using MRI. All patients were treated conservatively and underwent MRI after 3 months. Subsequent domino OVFs were defined as newly occurring OVFs within 3 months. Patient characteristics, types of conservative treatment, and patient-reported outcomes, including a visual analogue scale (VAS), the Oswestry disability index (ODI), and the Japanese Orthopaedic Association back pain evaluation questionnaire (JOABPEQ), were evaluated and compared between the domino OVF and non-domino OVF groups. *Results*: A total of 227 patients were analyzed. The mean age was 80.1 ± 7.3 years and 78% were female. Subsequent domino OVFs were observed in 31 (13.6%) patients within 3 months. An increasing number of prevalent OVFs were significantly associated with domino OVFs (*p* = 0.01). No significant differences in bone mineral density, type of brace, and anti-osteoporosis medications were found between the two groups. The JOABPEQ (excluding social function), ODI, and VAS were significantly improved after 3 months. Patients with domino OVFs at 3 months had poorer JOABPEQ social life function, ODI, and VAS than those with non-domino OVFs. *Conclusions*: In this study, the incidence of domino OVFs was 13.6% within 3 months. Domino OVFs had a negative impact on QOL at 3 months and were associated with prevalent OVFs.

## 1. Introduction

Fragility osteoporotic vertebral fractures (OVFs), the most common type of osteoporotic fracture, are increasing with the aging population and can increase morbidity and mortality. While the prognosis for most OVFs is favorable, there exist some cases where non-union, residual back pain, and the need for surgical intervention can arise. In Japan, the incidence of new cases of osteoporosis is 287.39 per 1000 person-years, and every year, approximately 970,000 people develop osteoporosis [1].

Recently, secondary fracture prevention has become important. It has been reported that the annual incidence of new vertebral fractures is 24.5 per 1000 person-years, while the incidence of subsequent vertebral fractures is higher at 68.8 per 1000 person-years [2,3]. Several studies have reported that the subsequent fracture risk is higher immediately following an initial fragility fracture than the risk of imminent fracture [4,5]. The incidence of imminent fracture following fragility fracture has been reported as 7.6% within 1 year and 11.6% in 2 years [4]. Generally, the risk of subsequent fracture is highest within 1 year following initial fracture and the relative risk is increased 5.3 fold [6]. The highest risk of subsequent major osteoporotic fracture has been reported within 6 months of index vertebral fracture, with a hazard ratio of 4.1 [7]. Clinical vertebral fracture has been reported as having the highest risk of subsequent fracture, with 14% and 26% at 1 and 2 years, respectively [4].

Concomitant OVFs have been proposed as at least two acute OVFs, with a reported incidence of 11–26% [8,9,10]. Similarly, domino OVFs have been named for the chain reaction of acute OVFs within a short period of time and are assumed to be rapidly developing multiple OVFs over a few months. The occurrence of domino OVFs may have an adverse effect on the mobility of patients and subsequently decrease their quality of life (QOL) [9]. Thus, it is imperative to investigate the impact of subsequent domino OVFs on the patient’s post-injury life.

Magnetic resonance imaging (MRI) is the most useful tool for detecting acute OVFs. Conservative treatment after acute osteoporotic vertebral fractures (AOVFs) typically improves lower back pain (LBP) within 3 months [11]. However, time course MRI studies after AOVFs have demonstrated that 97.7% of patients showed signal changes at 3 months [12], yet it is unclear whether domino OVFs occurred simultaneously or consecutively within a short period. Hence, we conducted a prospective study to investigate the extent of the development of subsequent domino OVFs using MRI at 3 months. This study aimed to assess the incidence of domino OVFs and their impact on QOL and residual LBP following conservative treatment for AOVFs.

## 2. Materials and Methods

### 2.1. Study Design and Patient Selection

A prospective, multicenter cohort study was conducted at eight hospitals between July 2020 and May 2022. This study complied with the tenets of the Declaration of Helsinki and was approved by the hospital’s institutional review board. Informed consent was obtained from all study participants. The inclusion criteria were (1) age > 60 years, (2) participation within 3 weeks from the initiation of the back pain, (3) AOVFs confirmed on plain radiographs and MRI, and (4) minor trauma or lack of significant trauma. The exclusion criteria were (1) vertebral tumors, (2) spine infections, (3) requirement of surgery due to neurological deficit or non-union, (4) previous spine surgery, (5) three or more prevalent OVFs, and (6) lack of radiographic data. Patient-related data and radiographic findings were obtained from electronic medical records.

### 2.2. Conservative Treatment

Conservative treatments for AOVFs include bed rest, physical therapy, bracing, and anti-osteoporosis medication. Medication for osteoporosis includes teriparatide, bisphosphonates, romosozumab, and denosumab, which were selected based on the severity of osteoporosis, medication adherence, and medical comorbidities. Anabolic agents, specifically teriparatide and romosozumab, are recommended for patients who have a high risk of fragility fractures. This includes individuals who have a prevalent OVF, T-scores lower than −2.5 standard deviations (SDs), and a semi-quantitative (SQ) grading of 3. The choice between teriparatide and romosozumab was made based on contraindications, including hypercalcemia, hypocalcemia, prior malignancy, and a history of stroke or myocardial infarction within the past year. Antiresorptive agents, including bisphosphonates and denosumab, were used when there was a contraindication for anabolic agents, the T-scores were higher than −2.5, or the patient was unable to perform self-injection due to advanced age. The selection of anti-osteoporosis medication was determined by the attending physician. Hospitalization was indicated for severe disability due to LBP. The type of brace was selected by physicians based on fracture type, patient age, brace compliance, and medical comorbidities. Braces were used for at least 3 months. Bone mineral density (BMD) was examined in all patients using dual-energy X-ray absorptiometry at the levels of the lumbar spine and hip. Baseline procollagen type 1 amino-terminal propeptide (P1NP) and tartrate-resistant acid phosphatase 5b (TRACP5b) levels were also evaluated.

### 2.3. Patient-Reported Outcome Measures (PROMs)

The Japanese Orthopaedic Association back pain evaluation questionnaire (JOABPEQ) was used to calculate pain-related disorders, lumbar function, walking ability, social life function, and mental health, yielding scores ranging from 0 to 100. Higher scores indicated better function. Visual analogue scale (VAS) scores for LBP ranged from 0 to 100, with higher scores indicating more severe pain. The Oswestry disability index (ODI) ranged from 0 to 100, with higher scores indicating poorer QOL. The questionnaires were completed at baseline and 3 months after conservative treatment.

### 2.4. Imaging Assessment

MRI was performed at baseline and at 3 months, using a 1.5 T MRI at each hospital. All fractures were diagnosed by six orthopedic surgeons (T.K., K.M., M.T., T.Y., M.H., and K.N.; orthopedic spine surgeons with 8, 21, 8, 7, 6, and 6 years of experience, respectively) using MRI. The morphology of vertebral collapses was classified into three types, depending on the site of the maximum reduction in vertebral height: wedge deformity, biconcave deformity, and crush deformity [13]. The SQ method was used to assess vertebral collapse according to four grades: 0 indicating “non-fracture,” 1 “mild fracture,” 2 “moderate fracture,” and 3 “severe fracture.” The categorizations of 0, 1, 2, and 3 correspond to reductions in vertebral height of ≤20%, 21–25%, 26–40%, and ≥41%, respectively. Subsequent domino OVFs were diagnosed using whole-spine sagittal MRI at 3 months.

### 2.5. Data Analysis

All measured variables are expressed as mean ± SD. Patient characteristics, types of conservative treatment, and outcome measures of the domino OVF and non-domino OVF groups were compared. We analyzed continuous variables using the Student’s *t*-test for normally distributed data. Continuous data with skewed distribution were analyzed using the Mann–Whitney U test after normality was assessed using the Shapiro–Wilk test. Fisher’s exact or chi-square tests were used to assess for categorical variables. Clinical outcomes, including JOABPEQ, VAS for LBP, and ODI at baseline and 3 months, were compared using the Wilcoxon signed-rank test. Statistical analyses were performed using JMP version 14 (SAS Institute, Cary, NC, USA). All tests were two-sided, and *p* values < 0.05 were considered significant.

## 3. Results

### 3.1. Patient Characteristics

A total of 277 patients who were conservatively treated for AOVFs were included in this study. Of these, 50 patients were excluded due to the following reasons: 13 patients died, 12 cases required surgery, 5 cases transferred to other hospitals, and 20 cases lacked complete data. Finally, 227 patients were analyzed. Subsequent domino OVFs were observed in 31 (13.6%) patients using MRI at 3 months (Figure 1). The mean age was 80.1 ± 7.3 years (range: 60–98 years). There were 177 females and 50 males. Prevalent OVFs were found in 44.9% (102) of patients (1 in 70 patients and 2 in 32 patients). Multiple AOVFs were observed in 40 patients (17.6%): 33 with two vertebrae, 5 with three vertebrae, and 2 with four vertebrae. The baseline characteristics of the patients, BMD data, and bone metabolism markers are presented in Table 1.

Regarding conservative treatment, 84.1% of patients did not receive anti-osteoporosis medication, 21 patients (9.3%) used bisphosphonates, 3 (1.3%) used teriparatide, 0 (0%) used romosozumab, 3 (1.3%) used denosumab, and 9 (4%) used selective estrogen receptor modulators at baseline. Out of all the patients, 82% (162 patients) exhibited severe osteoporosis, and of those patients, 83.3% (135 patients) were not undergoing osteoporosis treatment. After AOVFs treatment, 62 patients (27.3%) used bisphosphonates, 83 (36.6%) used teriparatide, 29 (12.8%) used romosozumab, 15 (6.6%) used denosumab, 6 (2.6%) used selective estrogen receptor modulators, and 32 (14%) did not receive any medication. A total of 49.3% of the patients used anabolic agents, including teriparatide and romosozumab. In all, 99 (43.6%) patients required hospitalization because of severe disability due to back pain. Furthermore, 106 patients (47.3%) used a soft brace, 110 (49.1%) used a rigid brace, and 11 (3.6%) used a support belt or no brace. The distributions of the prevalent, initial, and subsequent domino OVFs levels are shown in Figure 2.

At baseline, the morphology of the fractures was found to be 60.3% wedge deformity, 16.1% biconcave deformity, and 14.4% crush deformity. Three months after conservative treatment, the fracture morphologies were as follows: wedge deformity in 55.8%, biconcave deformity in 13.2%, and crush deformity in 27.4%. Regarding the degree of vertebral collapse, Grade 0 was observed in 9.2%, Grade 1 in 66.1%, Grade 2 in 22.4%, and Grade 3 in 2.3% of cases at baseline. After 3 months of conservative treatment, Grade 0 was observed in 3.6%, Grade 1 in 38.6%, Grade 2 in 35%, and Grade 3 in 22.8% of cases. Regarding the region of the spine, 23 OVFs (8.2%) were observed at the thoracic level (T5-9), 175 (62.7%) at the thoracolumbar level (T10-L2), and 81 (29%) at the lumbar level (L3-5) at the first visit. Prevalent OVFs were observed in 17 OVFs (12.5%) at the thoracic level, 75 (55.1%) at the thoracolumbar levels, and 44 (32.4%) at the lumbar levels. Subsequent domino OVFs were observed in 9 OVFs (12.8%) at the thoracic level, 24 (61.5%) at the thoracolumbar levels, and 10 (25.6%) at the lumbar level (Figure 3). There were no significant differences in the spinal region among the three groups. The proportion of domino OVFs was 84% (26 patients) in one vertebra, 10% (3 patients) in two vertebrae, and 6% (2 patients) in three vertebrae. There were 20 patients (51.3%) in the adjacent level and 11 patients (35.5%) in the remote level.

### 3.2. Comparison of Characteristics and Treatment between the Non-Domino OVF and Domino OVF Groups

There were no significant differences in patient demographics, including age, sex, height, body weight, and body mass index between the two groups (Table 2). An increasing number of prevalent OVFs were significantly associated with domino OVFs (*p* = 0.01). However, young adult mean values for the lumbar spine and total hip and Hounsfield units (HUs) at L4 were equivalent between the two groups (Table 2). TRACP5b and P1NP levels were also equivalent between the two groups. No significant differences were found in the type of brace, need for hospitalization, and type of anti-osteoporosis medication (Table 3).

Before the initial AOVF treatment, in the non-domino OVF group, 20 patients (10.2%) used bisphosphonates, 2 (1%) used teriparatide, 0 (0%) used romosozumab, 3 (1.5%) used denosumab, 8 (4.1%) used selective estrogen receptor modulators, and 163 (83.2%) did not receive any medication; and in the domino OVF group, 1 patient (3.2%) used bisphosphonates, 1 (3.2%) used teriparatide, 0 (0%) used romosozumab, 0 (0%) used denosumab, 1 (3.2%) used selective estrogen receptor modulators, and 28 (90.3%) did not receive any medication. After AOVF treatment, in the non-domino OVF group, 57 patients (29.1%) used bisphosphonates, 71 (36%) used teriparatide, 22 (11.2%) used romosozumab, 11 (5.6%) used denosumab, 6 (3.1%) used selective estrogen receptor modulators, and 29 (14.8%) did not receive any medication; and in the domino OVF group, 5 patients (16.1%) used bisphosphonates, 12 (38.7%) used teriparatide, 7 (22.6%) used romosozumab, 4 (12.9%) used denosumab, 0 (0%) used selective estrogen receptor modulators, and 3 (9.7%) did not receive any medication. There was no significant difference observed in the usage of osteoporosis medications between the two groups, both before and after the implementation of treatment intervention.

In all, 112 patients (49.3%) were treated for AOVFs with anabolic agents such as teriparatide and romosozumab. Anabolic agents were utilized for the treatment of osteoporosis in 93 patients (47.4%) in the non-domino OVF group and 19 patients (61.3%) in the domino OVF group, without any statistically significant difference between the two groups.

### 3.3. Comparison of PROMs between the Non-Domino OVF and Domino OVF Groups

PROMs were collected without missing data from 173 patients in the non-domino OVF group and 22 in the domino OVF group. There were no significant differences in baseline PROM scores between the non-domino and domino OVF groups. The social life function in the JOABPEQ at 3 months was significantly poorer in the domino OVF group than in the non-domino OVF group. The ODI and VAS for LBP were significantly higher in the domino OVF group than in the non-domino OVF group (*p* = 0.04 and *p* < 0.01). Severe disability (ODI > 40%) at 3 months was significantly greater in the domino OVF group than in the non-domino OVF group. All PROMs were significantly improved after 3 months, except for social function (*p* = 0.43) (Table 4, Figure 4).

## 4. Discussion

Subsequent domino OVFs is a condition in which multiple AOVFs are present because a new fracture develops before the initial OVF heals. In the present prospective study, 13.6% of the patients developed domino OVFs within 3 months. The incidence of subsequent OVFs after a new OVF was reported to be 6.7–19% (Inose, 6.7%; Yamauchi, 17.6%; and Lindsay, 19%) [3,8,14]. Inose et al. [3] reported that 6.7% of subsequent OVFs were observed within 48 weeks, and 73% of these occurred within 6 months.

Previous studies on conservative treatment did not use anabolic agents, such as teriparatide or romosozumab. Our study used anabolic agents in 49.3% of the patients. However, the incidence of domino OVFs was higher than that reported in previous studies. There were several reasons for this: (1) Our baseline population included multiple AOVFs (17.6%). These patients had a high risk of domino OVFs. (2) A total of 45% of patients had one or two prevalent OVFs. The risk of new OVFs has been reported to increase with the number of prevalent OVFs, which was 3.4 fold for one prevalent OVF and 7.4 fold for two or more prevalent OVFs [15]. (3) MRI was performed extensively on the thoracic spine to detect the fractures. Therefore, once AOVFs occur in patients with a high risk of subsequent OVFs, anti-osteoporosis medication alone cannot prevent domino OVFs within a short period of time.

Risk factors associated with subsequent OVFs after conservative treatment include LBP at baseline, type of brace, low lumbar BMD, worse functional recovery [8], and prevalent OVFs. Several studies have reported that prevalent OVFs were associated with the risk of subsequent OVFs [14,15]. Lindsay et al. [14] reported that the risk of subsequent OVFs increased by five-fold, which was consistent with our findings. Prevalent OVFs was the only predictor of domino OVFs in this study. Lumbar and femoral BMD, type of brace, and anti-osteoporosis medication were not found to be predictors of domino OVFs. The L4 HU value has been reported to be useful in predicting OVFs [16]. In this study, the L4 HU and BMD tended to be lower in the domino OVF group; however, the differences were not significant. Recent studies have reported that the combination of L1-L3 BMD and trabecular bone score are better tools for predicting OVFs [17]. Regarding mechanical factors, developing a wedge deformity increases the intensity of the flexor moment as a domino effect, resulting in rapid, multiple OVFs [18,19]. The authors concluded that T7, T8, T12, and L1 are critical vertebrae, as they are the inflection points of spinal alignment [18,19]. A recent finite element analysis showed that kyphotic deformation of the vertebral body at T12 increased the bimodal compressive stress on the adjacent vertebrae at T10 and T11 and midthoracic spine at T7 and T8 [20]. The presence of adjacent and remote dominoes in this study also indicated a difference in mechanical loading. Multiple factors, such as poor bone quality and mechanical stress, may impact the occurrence of domino OVFs.

Despite the possibility of an earlier onset of domino OVFs, assessing the clinical outcome at 1 month would be challenging as it would be difficult to discriminate between the source of pain arising from the initial OVF and that from the domino OVFs. Several recent studies have reported that pain and QOL significantly improve after 3 months of conservative treatment for OVFs [11,21,22]. Therefore, our study evaluated clinical outcomes and image assessments at the 3-month mark. We believe this timeframe provides a suitable interval for the natural course of conservative treatment following the initial OVF. In this study, the JOABPEQ, VAS for LBP, and ODI were significantly improved in both groups at 3 months. This study also included asymptomatic domino OVFs because all patients underwent MRI. However, the domino OVF group had higher VAS and ODI scores at 3 months and significantly worse social life function than the non-domino OVF group.

In a subgroup analysis of domino OVFs, patients with adjacent OVFs and number of OVFs ≥ 3 had significantly severe functional disability. Previous studies have reported that non-union, severe vertebral collapse, angular instability, and MRI findings, such as T2 high-signal changes, were associated with residual LBP [11,22,23]. Hu et al. reported that the number of OVFs was significantly correlated with global sagittal alignment, which was associated with a poorer QOL [24]. Although spinal alignment was not investigated in this study, short-term results were obtained, and care should be taken because an increasing number of OVFs is likely to result in worsened global spinal alignment.

In this study, the domino OVF group had significantly worse social life function, VAS for LBP, and ODI scores at 3 months compared to the non-domino OVF group. Although both groups had significant improvement in VAS for LBP and ODI after the 3-month follow up, social life function in the domino OVF group did not show significant improvement. It is noteworthy that the evaluation of walking ability in the study by Yamaura et al. [9] was conducted on a four-point scale, while the current study used the JOABEQ questionnaire to assess physical function, which may have contributed to the lack of significant difference in walking ability observed in this study. Furthermore, Ahmadi et al. [11] reported that LBP after vertebral fracture improved after 3 months of conservative treatment; thus, this study suggests that subsequent domino OVFs within the short-term period have a negative impact on VAS for LBP. The findings of this study suggest that domino OVFs may negatively impact the later QOL of patients. However, as the follow-up period was only 3 months, future studies are needed to assess the long-term clinical outcomes of domino OVFs.

The prevention of domino OVFs may play a key role in the success of conservative treatment for OVFs. In our study, only 16% of the patients received anti-osteoporosis treatment at baseline. In Japan, the number of new osteoporosis patients is increasing significantly each year, and consequently, the number of patients with vertebral fractures is also on the rise [1,2]. In recent years, guidelines for the prevention and treatment of osteoporosis have recommended that fragility OVFs be diagnosed as osteoporosis, regardless of BMD [25]. In addition, patients with fragility OVFs should be managed by a fracture liaison service (FLS) or multidisciplinary team, as recommended for secondary fracture prevention. Despite its high prevalence, a considerable number of patients with osteoporosis remain untreated; in this study, 84.1% of patients were not receiving any treatment. A total of 86% of our patients initiated anti-osteoporosis treatment after new OVFs, and 49.3% used anabolic agents. Several previous studies on the conservative management of OVFs did not incorporate anabolic agents, leading to a presumed lower incidence of subsequent domino OVFs in the current study [3,6,12,22,23]. However, the results of this study indicate that anti-osteoporosis treatment did not prevent domino OVFs. Domino OVFs may develop before the effects of anti-osteoporosis treatment, within a short period of time. While it is a widely recognized fact that drugs for osteoporosis are efficacious against fractures resulting from fragility, there exists a lack of clarity concerning the duration required for the onset of their effectiveness from the commencement of treatment. Bouxsein et al. [26] reported that a 2% improvement in lumbar BMD was associated with a 28% reduction in vertebral fracture, whereas 8% and 14% improvements in lumbar BMD were associated with 62% and 79% reductions in vertebral fracture, although the timing of this effect is uncertain since the rate of BMD increase varies depending on the medication used. It is advisable to initiate therapeutic intervention for osteoporosis as early as possible after diagnosis. Inose et al. [3] recommended the use of a rigid brace to prevent subsequent OVFs. However, the use of a rigid brace for all patients, such as the geriatric population, is uncommon. Therefore, decreasing the number of untreated patients with osteoporosis before the occurrence of new OVFs and increasing FLSs are considered the most effective approach for domino OVF prevention.

This study had several limitations. First, conservative treatment, including anti-osteoporosis treatment and type of brace, were selected based on the physician. Only two of all hospitals in this multicenter study used an FLS team. Second, we did not evaluate spinal alignment, the degree of vertebral collapse, and MRI findings. These factors may affect clinical outcomes and QOL scores. Third, 50 (18%) patients were excluded from this study. Twelve patients required surgery. The incidence of domino OVFs may underestimate the true number of patients. Fourth, a 3-month follow-up period may not provide adequate time to assess the clinical outcomes of domino OVFs. Longer-term monitoring is warranted, as this duration may not suffice to gauge the effectiveness of anti-osteoporosis medication and its impact on fracture prevention. However, the strength of this study was that the incidence of domino OVFs could be studied using MRI in a prospective multicenter manner.

## 5. Conclusions

In conclusion, the current study demonstrated that the incidence of subsequent domino OVFs was 13.6% within 3 months. Patients with domino OVFs had poorer social life function in the JOABPEQ, ODI, and VAS for LBP than in the non-domino OVF group. The types of anti-osteoporosis treatment and brace did not prevent domino OVFs. Detection and early intervention for severe osteoporosis before initial OVF may reduce domino OVFs.

## Figures and Tables

**Figure 1 medicina-59-00590-f001:**
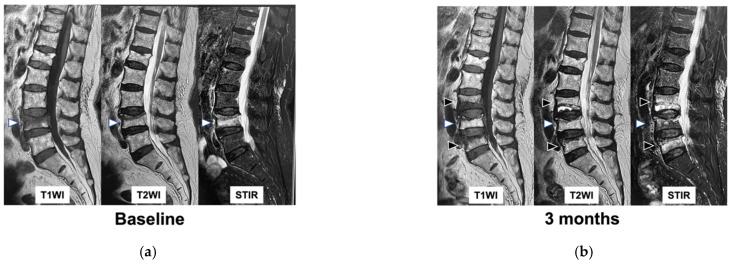
MRI findings of subsequent domino OVFs. (**a**): MRI image at baseline, (**b**): MRI image at 3 months. White arrows indicate the initial AOVF. Black arrows indicate subsequent domino OVFs. T1WI: T1-weighted imaging, T2WI: T2-weighted imaging, STIR: short τ inversion recovery, MRI: magnetic resonance imaging, OVFs: osteoporotic vertebral fractures, AOVF: acute osteoporotic vertebral fracture.

**Figure 2 medicina-59-00590-f002:**
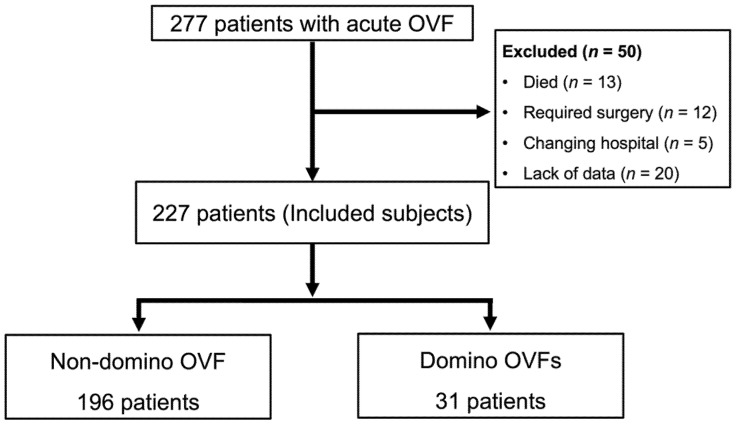
Schematic diagram for patient enrollment and follow up. A total of 277 patients treated conservatively for AOVFs were included in this study. Of these, 50 patients were excluded, and 227 patients were analyzed.

**Figure 3 medicina-59-00590-f003:**
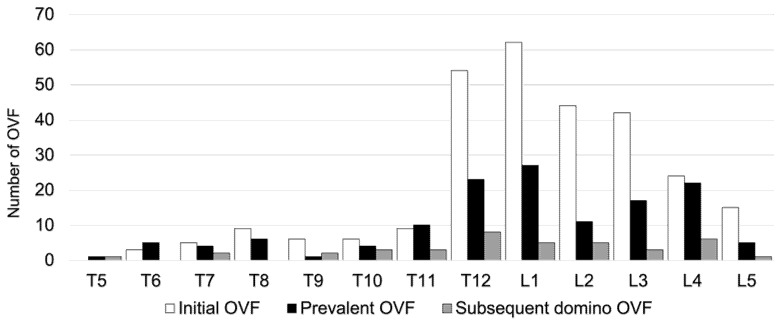
Distributions of the levels of the prevalent, initial, and subsequent domino OVFs.

**Figure 4 medicina-59-00590-f004:**
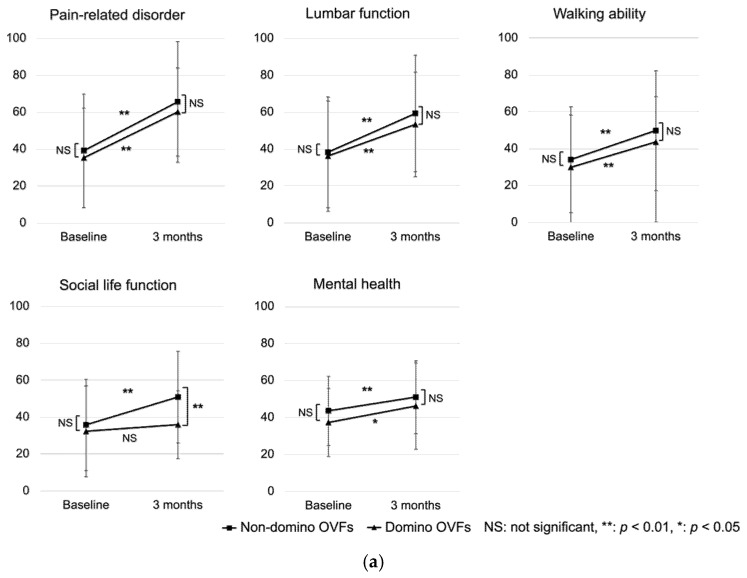
Result of PROMs at baseline and 3 months. (**a**): JOABPEQ, (**b**): VAS for LBP and ODI.

**Table 1 medicina-59-00590-t001:** Patient characteristics at baseline.

Variables	Total *n* = 227
Age (years)	80.1 ± 7.3
Sex (female, *n*, (%))	177 (78)
Weight (kg)	52.4 ± 9.5
Height (cm)	153.1 ± 9.5
BMI (kg/m^2^)	22.4 ± 3.6
Number of prevalent OVFs (*n*, (%))	
0	125 (55)
1	70 (31)
2	32 (14)
Number of initial AOVFs (*n*, (%))	
1	187 (82.4)
2	33 (14.5)
3	5 (2.2)
4	2 (0.8)
HU values (L4)	56.4 ± 33.8
Lumbar YAM (%)	77.3 ± 15.9
Femoral YAM (%)	70.2 ± 14.1
TRACP5b (mU/dL)	469.3 ± 218.4
P1NP (μg/L)	76 ± 76.1

BMI: body mass index, HU: Hounsfield unit, YAM: young adult mean, TRACP: tartrate-resistant acid phosphatase, P1NP: procollagen type 1 amino-terminal propeptide.

**Table 2 medicina-59-00590-t002:** Comparison of patient characteristics between the non-domino OVFs and the domino OVFs at baseline.

Characteristics	Non-domino OVFs *n* = 196	Domino OVFs *n* = 31	*p* Value
Age (years)	79.7 ± 7.5	82.1 ± 1.3	0.1
Sex (men/women)	41/155	9/22	0.31
Body weight (kg)	52.5 ± 9.7	51.9 ± 8.7	0.75
Height (cm)	152.8 ± 8	155.2 ± 9.7	0.16
Body mass index (kg/m^2^)	22.5 ± 3.7	21.5 ± 2.5	0.22
Number of prevalent OVFs (*n*, %)			0.01 *
none	113 (57.7)	12 (38.7)	
1	60 (30.6)	10 (32.2)	
2	23 (11.7)	9 (29)	
Multiple AOVFs	32 (16)	8 (26)	0.22
HU values (L4)	57.8 ± 33.7	47.9 ± 34.4	0.2
Lumbar BMD (YAM value, %)	77.7 ± 16.3	74.6 ± 12.7	0.41
Total hip BMD (YAM value, %)	70.6 ± 14.6	67.8 ± 10.3	0.34
TRACP5b (mU/dL)	458.7 ± 209.8	540.2 ± 262.7	0.09
P1NP (μg/L)	76.1 ± 73.4	75.7 ± 93.8	0.47

BMD: bone mineral density; * *p* < 0.05 indicates a statistically significant difference.

**Table 3 medicina-59-00590-t003:** Comparison of conservative treatment between the non-domino OVFs and the domino OVFs at baseline.

Variables	Non-domino OVFs *n* = 196	Domino OVFs *n* = 31	*p* Value
Type of brace (*n*, %)			0.1
hard	89 (45.4)	17 (54.8)	
soft	97 (49.5)	13 (41.9)	
none	7 (3.6)	1 (3.2)	
Hospitalization (yes (*n*, %))	86 (43.9)	13 (41.9)	0.84
Anti-osteoporosis medications (*n*, %)			
Bisphosphonate	57 (29.1)	5 (16.1)	0.54
SERMs	6 (3.1)	0 (0)	
Denosumab	11 (5.6)	4 (12.9)	
Teriparatide	71 (36)	12 (38.7)	
Romosozumab	22 (11.2)	7 (22.6)	
None	29 (14.8)	3 (9.7)	

SERMs: selective estrogen receptor modulators.

**Table 4 medicina-59-00590-t004:** Comparison of patient-reported outcomes between the non-domino OVFs and the domino OVFs at baseline and 3 months.

Measures	Non-domino OVFs *n* = 173	Domino OVFs *n* = 22	*p* Value
JOABPEQ			
Pain-related disorder			
At baseline	38 ± 30.6	36 ± 32.5	0.74
3 months	65.4 ± 32.1	59.9 ± 37.3	0.65
Lumbar function			
At baseline	33.9 ± 30.5	29.7 ± 27.8	0.51
3 months	59.1 ± 31.3	53.1 ± 33.9	0.35
Walking ability			
At baseline	30.7 ± 29.4	25.4 ± 23.7	0.52
3 months	49.6 ± 32	43.4 ± 36.4	0.31
Social life function			
At baseline	35.6 ± 24.4	32.1 ± 26.9	0.37
3 months	50.7 ± 24.7	35.7 ± 22.3	<0.01 *
Mental health			
At baseline	43.4 ± 18.3	37.1 ± 20.9	0.12
3 months	50.8 ± 19.1	46 ± 23.7	0.37
VAS low back pain			
At baseline	66.1 ± 25.6	68 ± 24.5	0.8
3 months	31.7 ± 25.1	43.7 ± 28.7	0.04 *
ODI			
At baseline	50.9 ± 22.7	54.5 ± 20.6	0.58
3 months	31.7 ± 20.6	44 ± 24.4	<0.01 *
ODI > 40% [*n*, (%)]			
At baseline	130 (69.1)	19 (73.1)	0.68
3 months	53 (30.3)	15 (60)	<0.01 *

JOABPEQ: Japanese Orthopaedic Association back pain evaluation questionnaire, VAS: visual analog scale, ODI: Oswestry disability index. * *p* < 0.05 indicates a statistically significant difference.

## Data Availability

The data presented in this study are available on request from the corresponding author. The data are not publicly available due to privacy and ethical restrictions.

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
