# Peer review of "Subsequent Domino Osteoporotic Vertebral Fractures Adversely Affect Short-Term Health-Related Quality of Life: A Prospective Multicenter Study"

_medicina, 2023, doi:10.3390/medicina59030590_

Round 1

Reviewer 1 Report

The author aimed to assess the incidence of domino OVFs after initial AOVFs, and their impact on quality of life. To my knowledge, anti-osteoporosis treatment is a very important part of conservative treatment (and Percutaneous Vertebroplasty is recommended for elderly patients to promote early activity rather than conservative treatment), but the choice of anti-osteoporosis medication and whether that choice improved the patients' osteoporosis, which would seriously affect the re-fracture rate (domino OVFs), was not discussed in this study. Also, the choice of 3 months for follow-up needs to be explained, because if the authors want to observe re-fracture they should choose a shorter period, such as 1 month, and if they want to see if the anti-osteoporosis medication improve the re-fracture rate then they should choose a longer period. Finally, the article has too many abbreviations and it is recommended to write the full name when it first appears.

Author Response

We thank you for your constructive comments and have attempted to address the suggestions and criticisms received.

Specifically:

Responses to Reviewer #1

Comment:

To my knowledge, anti-osteoporosis treatment is a very important part of conservative treatment (and Percutaneous Vertebroplasty is recommended for elderly patients to promote early activity rather than conservative treatment), but the choice of anti-osteoporosis medication and whether that choice improved the patients' osteoporosis, which would seriously affect the re-fracture rate (domino OVFs), was not discussed in this study.

Response:

As you pointed out, anti-osteoporosis treatment is a key role in conservative treatment for OVF. We have added the description of the choice of anti-osteoporosis medication in the method section.

We did not evaluate DXA at 3 months due to the lack of insurance coverage. We have added the impact of anti-osteoporotic medication on domino OVFs in the discussion.

Added sentence 1: (Page 2-3, line 93-104)

Anabolic agents, specifically teriparatide and romosozmab, are recommended for patients who have a high risk of fragility fractures. This includes individuals who have a prevalent OVF, T-scores lower than -2.5 standard deviations, and semi-quantitative (SQ) grading of 3. The choice between teriparatide and romosozmab was made based on contraindications, including hypercalcemia, hypocalcemia, prior malignancy, and history of stroke or myocardial infarction within the past year. Antiresorptive agents, including bisphosphonates and denosumab, were used when there was a contraindication for anabolic agents, the T-scores were higher than -2.5, or advanced age led to inability to perform self-injection. The selection of anti-osteoporosis medication was determined by the attending physician.

Added sentence 2: (Page 11, line335-338)

Several previous studies on conservative management of OVFs did not incorporate anabolic agents, leading to a presumed lower incidence of subsequent domino OVFs in the current study. [5, 9, 11, 19, 20] However, the results of this study indicated anti-osteoporosis treatment did not prevent domino OVFs. Domino OVFs may develop before the effects of anti-osteoporosis treatment, within a short period of time.

Comment:

 Also, the choice of 3 months for follow-up needs to be explained, because if the authors want to observe re-fracture they should choose a shorter period, such as 1 month, and if they want to see if the anti-osteoporosis medication improve the re-fracture rate then they should choose a longer period.

Response:

We have added a reason why 1 month is too short for clinical evaluation and added to the limitation that 3 months is too short to evaluate treatment of osteoporosis.

Added sentence:

(Page 10, line 289-291)

Despite the possibility of an earlier onset of domino osteoporotic vertebral fractures (OVFs). Assessing the clinical outcome at 1 month would be challenging, as it would be difficult to discriminate between the source of pain arising from the initial OVF and that from the domino OVFs. Therefore, our study evaluated clinical outcomes and image assessments at the 3-month mark. We believe this timeframe provides a suitable interval for the natural course of conservative treatment following the initial OVF.

(Page 11, line 358-361)

Fourth, a 3-month follow-up period may not provide adequate time to assess the clinical outcomes of domino OVFs. Longer-term monitoring is warranted, as this duration may not suffice to gauge the effectiveness of anti-osteoporosis medication and its impact on fracture prevention.

Comment:

Finally, the article has too many abbreviations and it is recommended to write the full name when it first appears.

Response:

As you pointed out, abbreviations have been corrected to write the full name.

The assistant editor suggested exceeding 4000 words out to me, so we made modifications to introduction, result, and discussion.

We would like to resubmit this paper for your consideration and we look forward to your response. Thank you in advance for your time and trouble.

Reviewer 2 Report

The study aimed to investigate the incidence and impact of domino osteoporotic vertebral fractures (OVFs) on quality of life (QOL) within three months of conservative treatment for acute 18 osteoporotic vertebral fractures (AOVFs). The statistical analysis part is well done: the tests used were chosen correctly, as they respect the nature of the variables. The only (minor) comments to be made as follows: - if possible, reduce the name of section 3.2 - what test did you use for gender, shown in table 2? being proportions, a test for equality of proportions should be used. If it has not already been done, state it in the text.

Author Response

We thank you for your constructive comments and have attempted to address the suggestions and criticisms received.

Specifically:

Responses to Reviewer #2

Comment:

 - if possible, reduce the name of section 3.2

Response:

As you suggested, we have reduced the name from “Comparison of characteristics and type of conservative treatment between the non-domino OVFs and domino OVFs groups” to “Comparison of Characteristics and Treatment Between the Non-domino OVFs and Domino OVFs Groups”.

Comment:

 - what test did you use for gender, shown in table 2? being proportions, a test for equality of proportions should be used. If it has not already been done, state it in the text.

Response:

We analyzed gender in table 2 using the chi-square test. We showed how to do this in section 2.5 (page 3, line 131-140).

The assistant editor suggested exceeding 4000 words out to me, so we made modifications to introduction, result, and discussion.

We would like to resubmit this paper for your consideration and we look forward to your response. Thank you in advance for your time and trouble.

Reviewer 3 Report

The authors present an interesting patient collective. The descriptive overview of the patients is detailed and scientifically comprehensible. There are no really surprising, especially new findings to be drawn from the review presented here. However, the problems with subsequent fractures in the treatment of osteoporotic sintering fractures are confirmed in a large patient collective. 

Please adjust the following:

 - To complete the data, however, I miss an overview of the fracture morphologies (e.g. according to AO Spine or OF classification).

 - The graphs in Figure 4 should be enlarged as a matter of urgency; in the current form in the manuscript, the details are hardly recognisable. In part a) the orthographic error in "walking ability" should be corrected. 

Author Response

We thank you for your constructive comments and have attempted to address the suggestions and criticisms received.

Specifically:

Responses to Reviewer #3

Comment:

 - To complete the data, however, I miss an overview of the fracture morphologies (e.g. according to AO Spine or OF classification).

Response:

As you suggested we have added the Genant classification of vertebral fractures at baseline.

Added sentence: (page 3, line 122-128)

The morphology of vertebral collapses was classified into three types, depending on the site of the maximum reduction in vertebral height: wedge deformity, biconcave deformity, and crush deformity. The semiquantitative (SQ) method was used to assess vertebral collapse according to four grades: 0 indicating “non-fracture,” 1 “mild fracture,” 2 “moderate fracture,” and 3 “severe fracture.” The categorizations of 0, 1, 2, and 3 correspond to reductions in vertebral height of ≤20%, 21-25%, 26-40%, and ≥41%, respectively.

Added sentence: (page 4, line 165-171)

At baseline, the morphology of the fractures was found to be 60.3% wedge deformity, 16.1% biconcave deformity, and 14.4% crush deformity. Three months after conservative treatment, the fracture morphologies were as follows: wedge deformity in 55.8%, biconcave deformity in 13.2%, and crush deformity in 27.4%. Regarding the degree of vertebral collapse, Grade 0 was observed in 9.2%, Grade 1 in 66.1%, Grade 2 in 22.4%, and Grade 3 in 2.3% of cases at baseline. After 3 months of conservative treatment, Grade 0 was observed in 3.6%, Grade 1 in 38.6%, Grade 2 in 35%, and Grade 3 in 22.8% of cases.

Comment:

 - The graphs in Figure 4 should be enlarged as a matter of urgency; in the current form in the manuscript, the details are hardly recognisable. In part a) the orthographic error in "walking ability" should be corrected.

Response:

We have corrected these items in figure 4 and Table 4, as you pointed out.

Corrected words:

From “Walling ability” to “walking ability”

The assistant editor suggested exceeding 4000 words out to me, so we made modifications to introduction, result, and discussion.

We would like to resubmit this paper for your consideration and we look forward to your response. Thank you in advance for your time and trouble.
